# Energy Expenditure and Maintenance Requirements in Non-Pregnant First-Parity Sows

**DOI:** 10.3390/ani14223276

**Published:** 2024-11-14

**Authors:** Ryan S. Samuel, Soenke Moehn, Ronald O. Ball

**Affiliations:** 1Department of Animal Science, South Dakota State University, Brookings, SD 57006, USA; ryan.samuel@sdstate.edu; 2Department of Agricultural, Food and Nutritional Science, University of Alberta, Edmonton, AB T6G 2P5, Canada

**Keywords:** sow, maintenance energy requirement, calorimetry, energy expenditure, heat production, feeding levels

## Abstract

This study investigated energy expenditure in five non-pregnant sows after their first parity. The sows were fed either 1.0 or 2.0 times the maintenance energy intake over subsequent weeks. When fed 1.0 times the maintenance energy intake, sows consumed 22.8 MJ/day, and lost weight, and their heat production exceeded energy intake. At 2.0 times maintenance, they consumed 44.9 MJ/day, gained 1292 g/day, and had a respiratory quotient above one, indicating fat storage. Heat production was higher during eating than immediately afterwards, regardless of meal size. The study suggests that modern sows have a maintenance energy requirement greater than previously reported.

## 1. Introduction

Optimizing sow body condition is essential for improving productivity and ensuring animal welfare. The energy requirements for sows have evolved significantly due to advances in genetics, nutrition, and management practices. Modern breeds are a result of constant genetic selection, which has resulted in greater growth rates and higher productive and reproductive capacity [1]. These genetic improvements have led to sows with leaner body composition, higher metabolic rates, and, thus, increased daily nutrient requirements [2]. Current feeding recommendations [3,4,5] may not be appropriate for optimum productivity of the modern, high-producing sow. Therefore, the objective of this study was to determine energy metabolism (heat production (HP) and body weight (BW) gain/loss) and actual measured values for maintenance energy of non-pregnant sows at two energy intakes.

## 2. Materials and Methods

### 2.1. Animals

Experimental procedures used in this study were approved by the University Animal Policy and Welfare Committee of the University of Alberta [6]. Five Large White/Landrace sows, non-pregnant after their 1st litter (BW: 174 ± 11 kg), were selected from a herd (Swine Research and Technology Centre (SRTC), Edmonton, AB, Canada). 

### 2.2. Respiration System

Two independent respiration chambers were constructed, each using a standard farrowing crate with a rear door for the animals to enter and exit the chamber [6]. Access to the animals was through a removable piece of plexiglass on top of and near the front of the chamber. Fresh water was provided ad libitum via a nipple drinker and feed could be provided directly into a trough through a feed tube, which was sealed when not in use. The chambers had a volume of 2 m^3^ and were airtight, except for two air inlets at the front. Negative pressure was induced in the chambers by rotary vane pumps (Gast Model 1023, Gast Manufacturing, Benton Harbor, MI, USA), thus drawing in fresh air (250 L/min) through the inlets, which was evenly distributed via pipes running the length of the chambers with holes drilled every 30 cm. Total air volume removed from the chambers was recorded by two independent AC630 gas meters (Canadian Meter, Edmonton, AB, Canada) and recorded manually every 30 min. A sub-sample of the total air volume was drawn by two separate vacuum pumps from the main air flow and fed under positive pressure at a flow of 150 mL/min to the gas analyzers (oxygen (O_2_)—S103; carbon dioxide (CO_2_)—S153; methane (CH_4_)—S127, Qubit Systems, Kingston, ON, Canada). The O_2_, CO_2_, and CH_4_ concentrations of the sample air were recorded in 60-s intervals as the average of 200 samples by data acquisition software (C409, Qubit Systems, Kingston, ON, Canada). The experiment was conducted in a temperature-controlled room kept at 20 ± 0.5 °C.

### 2.3. Diets, Feeding, and BW and Fecal Sample Collection

Sows were fed a diet based on wheat and barley with equal contributions of soybean and canola meals (Table 1). An indigestible marker, diatomaceous earth (Celite; Imerys Filtration Minerals, San Jose, CA, USA), was included at 10 g/kg of diet [7]. The diets were formulated to provide all nutrients needed by a pregnant sow of 175 kg body weight [3]. Feed allowance was calculated based on the estimated metabolizable energy (ME) content of the diet to provide the published recommended maintenance intake of 458 kJ ME/kg^0.75^ [8], which was taken to represent 1.0 times maintenance and then doubled to provide 2.0 times maintenance. Sows were fed for one week at each feeding level before respiration measurements. Feed intake was determined daily and BW weekly. Grab samples of feces were taken during the last 3 days at each feeding level.

Sows were fed twice daily, except for respiration days, when a frequent feeding protocol was implemented [6]. On respiration days, they were given half their daily rations in 16, half-hourly meals, followed in the afternoon by the remaining half of the daily allowance in 1 meal. Thus, several physiological states could be identified (Figure 1): ‘nibbling’ during the frequent feeding phase, ‘meal fed’ during the 1 h needed to consume the afternoon meal, ‘post-prandial’ from 2 h to 8 h after the afternoon meal, and ‘fasting’ for the remaining duration of the 24 h study. This feeding regimen was imposed to test the effects of different feeding strategies on energy metabolism and to derive relationships between measurements during the ‘nibbling’, ‘meal fed’, ‘post-prandial’, and ‘fasting’ physiological states and 24 h values.

### 2.4. Analyses

Feed and feces nitrogen (N) and carbon contents were analyzed by quantitative combustion (Leco Corporation, St. Joseph, MI, USA). Lipid, neutral detergent fiber (NDF), acid detergent fiber (ADF), and ash were analyzed according to AOAC [9]. Gross energy (GE) was determined by bomb calorimetry (Leco Corporation, St. Joseph, MI, USA). Acid insoluble ash (AIA) was analyzed according to McCarthy [7].

### 2.5. Calculation of Results

#### 2.5.1. Acid Insoluble Ash Content

Acid insoluble ash (AIA) content was calculated as
(1)$$\frac{\mathrm{Final}\;\mathrm{tube}\;\mathrm{weight}\;–\;\mathrm{Initial}\;\mathrm{tube}\;\mathrm{weight}}{\mathrm{Sample}\;\mathrm{weight}}\times 100=\%\;\mathrm{AIA}$$

#### 2.5.2. Digestibility

The apparent total tract digestibility (ATTD) of individual dietary components and energy was determined using diatomaceous earth as an indigestible marker and an analysis for AIA [7]. The following formula using the AIA contents of feed (AIA_feed_) and feces (AIA_feces_) and the nutrient content of feed (Nutrient_feed_) and feces (Nutrient_feces_) was used to calculate the nutrient digestibility of individual nutrients:(2)$$\mathrm{Nutrient}\;\mathrm{digestibility}\;\left(\%\right)=1-\frac{{\mathrm{AIA}}_{\mathrm{feed}}{\;*\;\mathrm{Nutrient}}_{\mathrm{feces}}}{{\mathrm{AIA}}_{\mathrm{feces}}{\;*\;\mathrm{Nutrient}}_{\mathrm{feed}}}$$

#### 2.5.3. Gas Consumed or Produced

Liters of gas (V_gas_) consumed (O_2_, CO_2_, CH_4_) or produced were calculated for each 30-min period, as follows:V_gas_ (L/30 min) = total air flow (L)/30 min ∗ (% gas room air − % gas test period)(3)

#### 2.5.4. Heat Production

Heat production was calculated according to [10]. The formula was abbreviated by omitting the urinary nitrogen term because planned attempts to collect urine were unsatisfactory; thus, results are not included [6]. The effect of the omission of urinary nitrogen (i.e., protein metabolism) was 1% for every 12.3% of the total energy that was derived from protein [11]. Therefore, the formula used to calculate HP from gas exchange was
HP = (16.175 × VO_2_) + (5.02 × VCO_2_) − (2.17 × VCH_4_)(4)
where VO_2_, VCO_2_, and VCH_4_ represent volumes (L) of O_2_ consumed and CO_2_ and CH_4_ produced, respectively. Daily HP was calculated as the summation of HP for every 30 min of the 24 h period.

#### 2.5.5. Maintenance Energy Requirements

The maintenance energy requirement (ME_m_) was calculated according to the formulae of [12]:energy retention (ER) = total ME intake − HP(5)

For sows in a positive energy balance,
(6)$${\mathrm{ME}}_{m}=\frac{\mathrm{total}\;\mathrm{ME}\;\mathrm{intake}\;-1.43\times \;\mathrm{total}\;\mathrm{ER}}{{\left(\mathrm{average}\;\mathrm{BW}\right)}^{0.75}}$$

And, for sows in a negative energy balance,
(7)$${\mathrm{ME}}_{m}=\frac{\mathrm{total}\;\mathrm{ME}\;\mathrm{intake}\;-0.80\times \;\mathrm{total}\;\mathrm{ER}}{{\left(\mathrm{average}\;\mathrm{BW}\right)}^{0.75}}$$

Efficiencies of energy utilization often vary due to the source, namely dietary sources versus body tissue. In this experiment, the value 1.43 is the reciprocal of the efficiency for the utilization of ME from dietary sources where the value of 70% was reported by [13]. Alternatively, the value 0.80 was used because energy from body tissue replaces ME with a greater efficiency compared to dietary sources, specifically 80% [13].

### 2.6. Statistical Evaluation

Data in tables are presented as least square means ± standard error of the mean (SEM), unless otherwise stated. A statistical analysis of variance was performed using PROC MIXED (SAS Inst. Inc., Cary, NC, USA). The feeding level was treated as a fixed effect to categorize the animals based on the level of feed, allowing the model to evaluate the influence on the parameters of interest (i.e., weight gain, nutrient digestibility, and energy retention). Considering animals as random variables accounts for variability between animals without attributing it to time-dependent factors. Model statements were tested using the Kenward–Roger degrees of freedom method. Least square means were compared using the ‘pdiff’ option. Significance was taken at *p* < 0.05, while 0.05 ≤ *p* < 0.1 was regarded as a tendency. A correlation analysis was used to compare the measured 24 h value of EE to the 24 h values extrapolated from the means from each physiological state.

## 3. Results

### 3.1. Indirect Calorimetry System

Firstly, the accuracy of the oxygen sensors and recovery of the system were tested by injecting a measured flow of nitrogen over a known time into the sealed respiration chambers [6]. The total injected nitrogen was calculated and compared to the reduction in measured O_2_ by the sensors. The efficiency of this test was 102.6 ± 0.6% for the O_2_ sensors over repeated (*n* = 6) measurements. The linearity of the response of the O_2_ analyzers was tested using five gasses of known O_2_ concentrations (20.94%, 20.42%, 19.69%, 18.99%, and 0%). The analyzers responded linearly (R^2^ > 0.998, coefficient of variation (CV ) < 0.35%) to the series of gasses with O_2_ contents between 0 and 21%.

Secondly, the CO_2_ sensors and the entire system were tested by dissolving a known quantity of NaHCO_3_ into water. A solution was placed into each of the respiration chambers and concentrated HCl_(aq)_ was delivered from a syringe driven by an injection pump. The chambers were sealed before the syringe pumps were started. As the HCl_(aq)_ was delivered, CO_2_ evolved. The amount of CO_2_ measured by the sensors was quantified and compared to the quantity of NaHCO_3_ added. The recovery of CO_2_ was 107.3 ± 2.1% over repeated (*n* = 15) measures for both chambers, with no difference between chambers (*p* > 0.80). The CO_2_ analyzers’ response to changing CO_2_ content of the air was a non-linear response curve. The curve was repeatedly (*n* = 5) determined using seven gasses of known CO_2_ concentration (0, 0.04, 0.08, 0.4, 0.8, 1.2, and 1.5%). 

### 3.2. Energy Metabolism

Daily feed intake was 1.84 ± 0.03 kg for sows (BW: 175.7 ± 3.3 kg) fed to represent the estimate for 1.0 times maintenance and then doubled to 3.69 ± 0.05 kg to provide sows (BW: 174.1 ± 3.1 kg) an estimated intake of 2.0 times maintenance [8]. Actual DE energy intakes were calculated from energy digestibility data derived from the AIA analysis of feed and feces (Table 2), and ME intakes were calculated as 96% of DE [3]. Therefore, sows were fed 473 ± 5.5 kJ ME/kg^0.75^ and 925 ± 6.1 kJ/kg^0.75^ in the first and second part of the experiment, respectively. Mean daily gain was −198 ± 96 g/d and +1292 ± 215 g/d for sows fed 473 ± 5.5 kJ ME/kg^0.75^ and 925 ± 6.1 kJ/kg^0.75^, respectively. Nutrient digestibility was similar for both feeding levels, except for greater ATTD of ADF (*p* = 0.03) and a tendency for greater digestibility of N and OM (*p* ≤ 0.09) for the higher feeding level (Table 2).

Sows fed 473 ± 5.5 kJ ME/BW^0.75^ began the experiment weighing 175.7 ± 3.3 kg BW and ended the experiment at 174.1 ± 3.2 kg. However, the difference (−1.6 kg) was not significant (*p* = 0.11). Sows fed 925 ± 6.1 kJ/BW^0.75^ started at 174.1 ± 3.1 kg BW and gained (*p* < 0.0001) weight up to 186.6 ± 3.3 kg at the rate of 1292 ± 215 g/d.

Sows fed 473 kJ ME/kg^0.75^ had an average daily HP of 24.0 ± 0.4 MJ or 525 ± 9 kJ/kg^0.75^ (Table 3). Sows fed 925 kJ ME/kg^0.75^ had an average daily HP of 32.3 ± 0.5 MJ or 624 ± 10 kJ/kg^0.75^, which was significantly greater (*p* < 0.001) than sows fed at 473 kJ/kg^0.75^. Among sows receiving the same feeding level, HP was not different (*p* > 0.15).

### 3.3. Comparison to 24 h Values

There were no significant interactions between the feeding level and the ‘nibbling’, ‘meal fed’, ‘post-prandial’, or ‘fasting’ physiological states (*p* > 0.1); therefore, 24 h means from each physiological state were calculated independent of the feeding level (Table 4) [6]. Sows had significantly greater heat production (*p* = 0.03) with ‘nibbling’ or ‘meal fed’ compared to the ‘post-prandial’ and ‘fasting’ states [14]. The RQ, with ‘post-prandial’, was greater (*p* = 0.04) then during the ‘nibbling’ and ‘fasting’ phases.

The HP measured during physiological states was extrapolated to 24 h and compared to the 24 h measured HP of 28.1 MJ/d (Table 5) [6]. During the ‘nibbling’ phase, extrapolated HP exceeded the measured daily HP by a mean factor of 1.17 (*p* < 0.001) with no difference between feeding levels (*p* > 0.65). During ‘post-prandial’ and ‘fasting’, measured daily HP exceeded the values extrapolated from those physiological states. The measured daily HP was not different compared to the values extrapolated from ‘meal fed’ sows.

## 4. Discussion

### 4.1. Indirect Calorimetry System

Custom-built respiration chambers were combined with individual gas analyzers to create a pair of independent indirect calorimetry systems for measurement of energy metabolism [6]. Responses of the gas analyzers were validated; the O_2_ analyzers responded linearly and the CO_2_ analyzers responded in a predictable non-linear manner to changing gas concentrations. The gas analyzers responded accurately in simulated measurements of energy metabolism, as shown by recoveries of 100% of the predicted volume of O_2_ reduction or CO_2_ release. 

### 4.2. Energy Metabolism

The determined ME_m_ for each individual non-pregnant first-parity animal within this experiment was calculated using the formulae of Lodge [12]. For sows fed 473 kJ ME/kg^0.75^, the mean calculated ME_m_ was 515 ± 8 kJ/kg^0.75^ and for sows fed 925 kJ ME/kg^0.75^, the mean calculated ME_m_ was 495 ± 9 kJ/kg^0.75^. These calculated ME_m_ values are comparable to previous reports; namely, 550 kJ/kg^0.75^ was the calculated thermoneutral ME_m_ of early weaned piglets [15]. The calculated ME_m_ value for sows fed at an approximate ME_m_ intake was numerically greater (by 20 kJ) than for sows fed at approximately twice an ME_m_ intake. This phenomenon was previously demonstrated by Wenk [16], who reported a lower ME_m_ for ad libitum-fed animals than restrictively fed animals and concluded that this was due to reduced physical activity due to satiety. Halter [17] reported the energy requirement for activity to be 30% higher for restrictively fed piglets compared to ad libitum-fed piglets. The result of reduced physical activity diminishes HP and therefore the calculated ME_m_ is lower. The calculated ME_m_ values do not represent ad libitum feed intake because the feed intake of the sows was restricted to below ad libitum intake, even at 925 kJ/kg^0.75^ (approximately twice ME_m_). The calculated ME_m_ values from this experiment were not significantly different (*p* = 0.15). Therefore, the arithmetic mean of 506 ± 7 kJ/kg^0.75^ was taken as the ME_m_ for this population of modern sows. This is equivalent to 24.3 MJ /d for sows weighing 175 kg.

The mean ME_m_ value from this experiment was greater than the values suggested by ARC [8] of 458 kJ ME/kg^0.75^ and by NRC (1998) of 106 kcal ME/kg^0.75^ (equivalent to 444 kJ ME/kg^0.75^). This could be expected, because sow body weight and lean tissue mass increased and both raise maintenance energy and protein requirements due to the continuous process of protein turnover [18,19]. Protein turnover requires energy and amino acids, thus contributing to daily energy and amino acid requirements. The selection of pigs for greater leanness has increased body protein content, resulting in higher maintenance requirements [20]. Wenk [16] previously reported that ME_m_ values have increased for swine due to the increase in lean tissue turnover. 

### 4.3. Nutrient Metabolism

Mean ME intake from the diet was lower than HP and maintenance EE for sows fed 473 kJ ME/kg^0.75^ (Table 4); consequently, energy retention was negative (−0.1 MJ/d). This is consistent with the observed numerically negative weight gain (−198 ± 96 g/d) of the sows. 

Dietary energy intake exceeded HP and maintenance EE for sows fed 925 kJ ME/kg^0.75^, leading to positive energy retention (12.6 MJ/d). Sows gained 1292 g/d and, in agreement with the observed mean daily respiratory quotient (RQ) of greater than 1 (1.16 vs. 1.03), there was a significant gain of body lipids. 

The feed consumption of 3.69 kg/d provided 23.97 g/d of lysine when 2.0 times ME_m_ was fed. This approximates 19.18 g/d SID lysine, or 16.82 g/d SID lysine above the lysine maintenance requirement of 49 mg/kg^0.75^ [21]. When 12 g SID lysine is needed per 100 g protein deposition (PD), this intake could support PD of 140 g/d [3,22]. Based on an energy cost of the protein accretion of 44.4 kJ ME/g [3], 6.2 MJ ME would be utilized for a protein accretion of 140 g/d. Total daily ME intake of 44.9 MJ less the energy for ME_m_ (180 kg * 0.506 MJ/kg^0.75^) of 24.9 MJ ME leaves 20.0 MJ available for growth. Subtracting the energy cost of a protein accretion of 6.2 MJ from the energy available for growth (20.0 MJ) leaves 13.8 MJ ME per day available for fat accretion. Because the energy cost of fat accretion is 52.3 kJ/g of fat, this excess energy (13.8 MJ) would result in 263.9 g/d of fat accretion. The potential of 140 g/d protein and 254 g/d fat equates to a daily gain of 814 g/d, assuming the water content of deposited protein as 75% [23] and the water content of deposited fat is 10% [24]. The difference in this predicted gain to the measured average daily gain of 1292 g/d may be explained by an increase in gut fill caused by the increase in feed intake from 1.0 to 2.0 ME_m_. For example, Dourmad et al. [25] observed that both the empty digestive tract and digestive contents’ weights increased by ~600 and ~1200 g, respectively, as feed intake increased through lactation. Therefore, it is not unreasonable to attribute the 480 g of unaccounted weight gain in the comparison of predicted versus observed values to gastrointestinal tract growth and gut fill.

The mean digestibility of nutrients and energy in this experiment was similar to that reported by Noblet and Henry [26] and the calculated values based on data by NRC [3], but lower than values reported by Noblet and Shi [27]. The differences in digestibility from Noblet and Shi [27] compared to current results may be due to differences among Canadian and European wheat and barley cultivars. Fairbairn et al. [28] reported digestibility values for barley that were lower than those reported by Noblet and Shi [27] for European varieties. 

The observed increase in the ATTD of ADF and the tendency of increased ATTD of N and OM may be due to the fact that larger meals, such as feeding 2.0 times maintenance, increased feed retention time in the gastrointestinal tract potentially due to decreased motility and slower gastric emptying and would allow more time for microbial fermentation compared to smaller meals (1.0 times maintenance) [29]. However, measurement of nutrient digestibility was not the primary focus of the trial and results should be interpreted with caution.

### 4.4. Comparison to 24 h Values

The 24 h HP of sows was affected by the different feeding frequencies in a similar manner for both of the feeding levels. When sows were consuming small meals every 30 min, HP was greater (*p* < 0.01) than when sows were fasting or following the large meal (post-prandial). The HP of the sows while eating, whether consuming the frequent, small meals or consuming the single large meal, was not different. A portion of the greater HP associated with eating is due to the physical activity of sows when consuming a meal. Sows tend to be quite enthusiastic about feeding and will consume their meals rapidly while standing. Furthermore, sows displayed physical activity the morning following the overnight ‘fasting’ period, assumingly awaiting their morning meal. The other portion of the greater HP associated with eating is due to the heat increment of feeding (HIF). Thus, greater HP is due, in part, to the EE associated with standing and consuming the meal and, on the other hand, due to the digestion of the consumed meal [30]. Increasing feeding frequency may improve the efficiency of protein metabolism because the delivery of amino acids is more gradual [31,32]. 

Estimating daily energy expenditure is essential for understanding metabolic needs and guiding dietary recommendations in both animal and human studies. However, directly measuring energy expenditure over a full 24-h period can be challenging [33], time consuming, and costly [34]. To mitigate this, researchers often extrapolate shorter measurement intervals to estimate total daily energy expenditure, though this approach requires careful methodology to ensure accuracy. Statistical modeling and regression techniques have been successfully used to extrapolate shorter measurements by adjusting for specific time-dependent factors, such as rest versus activity periods [35]. These approaches provide a more reliable estimate by accounting for metabolic fluctuations that occur within a day. In livestock management, where continuous monitoring is impractical, extrapolating shorter measurement periods with these models offers a feasible and efficient method to estimate daily energy needs, aiding in more accurate feeding strategies [36]. The ability to achieve valid 24 h values from studies of a shorter duration [37] was studied by testing a frequent, half-hourly feeding regimen, modified from Moehn et al. [38,39], for 8 h followed by a bolus meal. Results from sows fed 473 and 925 kJ ME/kg^0.75^ indicate that 24 h values of energy metabolism can be accurately extrapolated from measurement periods of a shorter duration and during frequent feeding. Although the 24 h values of HP extrapolated from frequent feeding exceeded the measured daily HP (33.1 vs. 28.1 MJ/d), the effect was consistent enough across the sows and feeding levels investigated in this experiment to establish a relationship (i.e., fraction of 24 h HP). Furthermore, as suggested by Moehn et al. [38,39], frequent feeding is the best condition for indicator amino acid oxidation and energy metabolism studies due to the stability of carbon dioxide values, being equally valid with respect to daily HP and RQ measurements.

## 5. Conclusions

Independent respiration chambers were constructed around standard farrowing crates and combined with individual gas analyzers to create indirect calorimetry systems. Responses of the gas analyzers were validated. Results from this experiment indicate that previously reported ME_m_ values for non-pregnant sows are too low for modern breeds. Furthermore, these results indicate that maintenance energy needs of sows should be measured on a regular basis because sows have been and are rapidly changing due to genetic selection for increased litter size. These results also demonstrate that 24 h values of energy metabolism from measurement periods of a shorter duration and during frequent feeding can be utilized in future experiments.

## Figures and Tables

**Figure 1 animals-14-03276-f001:**
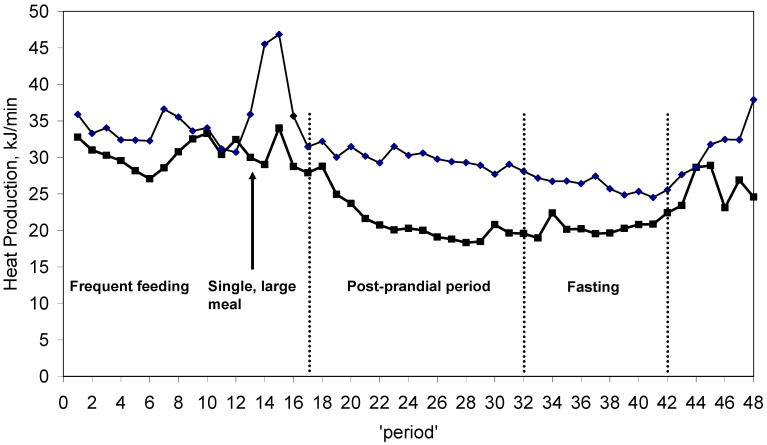
Mean heat production of sows fed 1.0 times maintenance (-□-) or 2.0 times maintenance (-◊-) according to collection ‘period’ and feeding physiological states ^1^. ^1^ Frequent feeding = 1/32 of daily feed allowance every 30 min; single, large meal = ½ of daily feed allowance after 8 h; post-prandial = starting 2 h after large meal and for 8 h; fasting = 10 h after large meal and for 6 h.

**Table 1 animals-14-03276-t001:** Diet composition of dry sow rations.

Ingredient	% of Diet
Wheat, 12.5% CP ^1^	12.1
Barley, 11.2% CP	69.0
Soybean meal, 48% CP	6.5
Canola meal, 35% CP	6.5
Canola oil	0.9
Breeder premix ^2^	4.0
Celite ^3^	1.0
**Calculated nutrients**	
Ca, %	0.95
Total P, %	0.72
DE ^4^, MJ/kg	13.06
CP, %	14.76
Total Lys, %	0.65

^1^ Crude protein. ^2^ Provided per kilogram of the diet: Ca, 8.6 g; P, 3.4 g; Na, 1.9 g; Mg, 140 mg; K, 30 mg; Fe, 139 mg; Zn, 119 mg; Mn, 56 mg; Cu, 16 mg; Co, 0.4 mg; I, 0.4 mg; Se, 0.3 mg; vitamin A, 12,000 IU; vitamin D3, 1200 IU; vitamin E, 62 IU; vitamin K, 2.5 mg; biotin, 0.6 mg; folic acid, 2.5 mg; niacin, 42 mg; pantothenic acid, 25 mg; pyridoxine, 5 mg; riboflavin, 9.5 mg; thiamine, 8.4 mg; vitamin B12, 28 μg. ^3^ Imerys Filtration Minerals, San Jose, CA, USA. ^4^ Digestible energy.

**Table 2 animals-14-03276-t002:** Effect of feeding level 1.0 times maintenance or 2.0 times maintenance on nutrient digestibility by sows.

Nutrient Digestibility, %	1.0 Times	2.0 Times	SEM	*p*-Value
Energy	79.7	80.9	0.8	0.34
Nitrogen	81.3 ^y^	83.6 ^x^	1.7	0.08
Carbon	81.6	82.3	1.2	0.74
Fat	87.3	85.6	3.0	0.73
NDF	42.3	61.2	5.9	0.11
ADF	24.4 ^b^	50.6 ^a^	4.9	0.03
Organic matter	84.1 ^y^	86.1 ^x^	0.9	0.09

^a,b^ Values with different superscripts in rows differ, *p* < 0.05. ^x,y^ Values with different superscripts in rows tend to differ, 0.05 ≤ *p* < 0.10.

**Table 3 animals-14-03276-t003:** Parameters of energy metabolism of five sows fed 473 or 925 kJ ME/kg^0.75^.

Energy Metabolism	1.0 Times	2.0 Times	SEM	*p*-Value
ME intake, MJ/d	23.9 ^b^	44.9 ^a^	3.8	<0.01
HP, MJ/d	24.0 ^b^	32.3 ^a^	1.5	0.04
Maintenance EE, MJ/d	24.3 ^b^	25.6 ^a^	0.3	<0.01
Energy retention, MJ/d	−0.1 ^b^	12.6 ^a^	3.1	<0.01
RQ	1.03 ^b^	1.16 ^a^	0.02	<0.01

^a,b^ Values with different superscripts in rows differ, *p* < 0.05.

**Table 4 animals-14-03276-t004:** Effect of feeding level and physiological state of sows on parameters of energy metabolism of sows extrapolated to 24 h.

	Nibbling	Meal Fed	Post-Prandial	Fasting	SEM	*p*-Value
Observations	10	8	8	8		
473 kJ ME						
HP, MJ/d	30.4 ^a^	27.9 ^ab^	20.9 ^c^	23.9 ^b^	2.6	<0.01
RQ	0.96 ^b^	1.08 ^ab^	1.15 ^a^	0.91 ^b^	0.07	<0.01
925 kJ ME						
HP, MJ/d	35.6 ^a^	31.5 ^ab^	29.4 ^c^	29.1 ^c^	2.6	<0.01
RQ	1.12 ^ab^	1.21 ^ab^	1.25 ^a^	1.09 ^b^	0.07	<0.05

^a,b,c^ Values with different superscripts in rows differ, *p* < 0.05.

**Table 5 animals-14-03276-t005:** Comparison of HP measured during identified physiological states extrapolated to 24 h daily measured HP.

	Nibbling	Meal Fed	Post-Prandial	Fasting	SEM	*p*-Value
Fraction of 24 h heat production	1.17	1.00	0.90	0.95	0.8	0.03

## Data Availability

The original contributions presented in the study are included in the article and further inquiries can be directed to the corresponding author.

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
