# Peer review of "Energy Expenditure and Maintenance Requirements in Non-Pregnant First-Parity Sows"

_animals, 2024, doi:10.3390/ani14223276_

Round 1
Reviewer 1 Report
Comments and Suggestions for Authors
In the study of “energy expenditure and maintenance requirements in first-parity sows”, Samuel and colleagues explored the MEm value in modern breed of sows during gestation and lactation, which revealed that MEm needs of sows should be measured on a regular basis in order to accommodate genetic selection for high reproductive performance. However, there are some concerns should be addressed.
Major concerns:
1. Limited Sample Size: The experimental sample size may be insufficient, particularly if based on a small number of first-parity sows, which could limit the generalizability of the results.
2. Statistical Analysis Clarity: The statistical methods used lacks a clear explanation of the statistical model and significance levels, making it difficult for readers to assess the robustness of the findings. This should be clarified in the section of Statistical Analysis.
3. Influence of Environmental Factors: The study did not adequately consider the impact of environmental factors (e.g., temperature, humidity) on energy metabolism, which can significantly affect energy expenditure in sows, especially in extreme conditions. This should be at least discussed.
4. Insufficient Data Supporting Conclusions: The study lacks sufficient statistical data to support the conclusions, particularly regarding energy requirements during different stages (early, middle, and late pregnancy, as well as lactation). More detailed quantitative data is needed to validate the findings.
5. Scope of Application: The discussion should explicitly state the conditions under which the conclusions apply to avoid inappropriate extrapolation and address study limitations such as sample size and environmental controls.
6. Conclusion should be more concise.
Minor concerns:
1. Table 1: The protein content of the raw material can be indicated next to it, e.g. Wheat, 12.5% CP. At the same time, it is better to list the percentage of each ingredient
2. The P for “P-value” should be italic.
Author Response
In the study of “energy expenditure and maintenance requirements in first-parity sows”, Samuel and colleagues explored the MEm value in modern breed of sows during gestation and lactation, which revealed that MEm needs of sows should be measured on a regular basis in order to accommodate genetic selection for high reproductive performance. However, there are some concerns should be addressed.
Response: The authors thank the reviewer for the detailed consideration of the paper. Changes are indicated below and marked in red in the updated manuscript
Comment 1: Limited Sample Size: The experimental sample size may be insufficient, particularly if based on a small number of first-parity sows, which could limit the generalizability of the results.
Response 1: The authors agree that the limited sample size is a concern for the generalizability of the results beyond the scope of this manuscript. However, it should be noted that the results within the project were consistent with each other and support the observations for this population of sows.
Comment 2: Statistical Analysis Clarity: The statistical methods used lacks a clear explanation of the statistical model and significance levels, making it difficult for readers to assess the robustness of the findings. This should be clarified in the section of Statistical Analysis.
Response 2: The experimental model investigated the effect of ‘feeding level’ as the classification variable on the parameters of interest. Individual animals were ‘random variables’ rather than utilizing repeated measures approach. A repeated measures approach would model each animal’s response over multiple time points, focusing on within-subject changes over time. Instead, by not using the repeated measures approach, the model is more likely examining the effect of feeding level on each animal independently. Significance was taken at P < 0.05, while 0.05 < P < 0.1 was regarded as a tendency. Lines 176-179 were updated: Feeding level was treated as a fixed effect to categorize the animals based on the level of feed allowing the model to evaluate the influence on the parameters of interest (i.e., weight gain, nutrient digestibility, and energy retention). Considering animals as random variables accounts for variability between animals without attributing it to time-dependent factors.
Comment 3: Influence of Environmental Factors: The study did not adequately consider the impact of environmental factors (e.g., temperature, humidity) on energy metabolism, which can significantly affect energy expenditure in sows, especially in extreme conditions. This should be at least discussed.
Response 3: The authors did not choose to include any discussion related to environmental factors within the manuscript because the experiments were conducted in a temperature controlled. Line 63 has been added “The experiment was conducted in a temperature controlled room kept at 20 ± 0.5 ⁰C.
Comment 4: Insufficient Data Supporting Conclusions: The study lacks sufficient statistical data to support the conclusions, particularly regarding energy requirements during different stages (early, middle, and late pregnancy, as well as lactation). More detailed quantitative data is needed to validate the findings.
Response 4: The authors agree with the statement. Thus, conclusions were narrowed for this manuscript by removing “Additionally, the continued selection for increased growth and efficiency of the offspring also results in metabolic changes in the dams”.
Comment 5: Scope of Application: The discussion should explicitly state the conditions under which the conclusions apply to avoid inappropriate extrapolation and address study limitations such as sample size and environmental controls.
Response 5: The authors have updated the first line of discussion to include “non-pregnant first-parity” and “within this experiment”.
Comment 6: Conclusion should be more concise.
Response 6: The authors have updated the Conclusion by removing “Additionally, the continued selection for increased growth and efficiency of the offspring also results in metabolic changes in the dams” and adding “non-pregnant” in Line 343.
Comment 7: Table 1: The protein content of the raw material can be indicated next to it, e.g. Wheat, 12.5% CP. At the same time, it is better to list the percentage of each ingredient
Response 7: The authors have updated the Table 1 as suggested.
Comment 8: The P for “P-value” should be italic.
Response 8: The authors have updated the manuscript as suggested.
Reviewer 2 Report
Comments and Suggestions for Authors
Comments.
The manuscript presents valuable research on energy expenditure and maintenance requirements in first-parity sows. The experimental design, particularly the use of indirect calorimetry and a well-controlled feeding regimen, provides important insights into the energy metabolism of modern sows. The findings regarding the higher maintenance energy requirements (MEm) in modern breeds compared to previously reported values are relevant and timely, given recent genetic advances. The study fills an important gap in understanding the energy requirements of sows, which is essential for optimizing productivity and welfare in the swine industry. In general, I would recommend that this manuscript be published in the journal, provided the authors address the following points.
1. The focus on first-parity sows and the detailed examination of energy expenditure at different feeding levels addresses a critical topic in swine nutrition, particularly in relation to modern genetics. One major flaw is that the current findings was obained using non-pregnant sows, yet most of the comparation was conducted against pregnant sows. It is a common concept that the pregnant sows had a greater energy utilization due to the pregnancy anobolism, thus the conclusion must be drawn carefully.
2. Methodology: The use of indirect calorimetry and frequent feeding protocols, combined with robust statistical analysis, ensures the reliability of the results. However, the use of only five (simetimes showed six) sows limits the generalizability of the findings. A larger sample size would strengthen the conclusions drawn. Additional, the number of animals was not consisitent througout the study. There are six sows described in line 43, yet only five sows were used in line 15.
3. In the statistical analysis, I am unsure how the data were evaluated. Was a repeated measures approach used, incorporating feeding level, period, and the interaction between feeding level and period?
4. In the Calculations of Energy Retention: The methodology for calculating maintenance energy requirements is solid, but the rationale for using different efficiency values (1.43 vs. 0.80) could be better explained, particularly for readers less familiar with this calculation model.
5. Statistical Power: While some differences are noted as tendencies (P values between 0.05 and 0.1), these should be interpreted with caution. It may be helpful to further discuss the biological relevance of these trends.
6. The manuscript could benefit from a clearer presentation of the key findings. For example, the figures and tables could be more intuitive by highlighting the most important comparisons and including more concise legends if possible. Also, for the Physiological States Analysis: Although the study examines various physiological states (e.g., 'nibbling', 'post-prandial'), the extrapolation of these states to 24-hour values is not entirely convincing. Additional explanation or references to justify these extrapolations would be beneficial.
7. Terminology: There are instances where more precise language could improve readability, especially when describing metabolic rates and energy turnover.
8. The discussion section should be improved.
Line 234-235, how MEm been determined by “using the formulae of 234 Lodge [11] (Table 5)”. Line 237, the ref. [14] was based on weaning piglets, could be suitable for elucidating this issue?
Line 240, indicates the animal (growing pig) when citing the ref. [15].
Line 262-263, I do not agree with the notion that a negative in energy retention (-0.1MJ/d) is consistent with a negative weight gain (-198 ± 96 g/d), that does not make sense. The RQ was above 1.0 (1.03) for the 1.0 time in table 3, indicating lipid retention. How to explain this?
I would suggest the authors discussion why the ADF and organic matter greater in 2.0 times than the 1.0 times.
9. Conclusion: This manuscript contributes significantly to the understanding of energy metabolism in modern sows. However, to enhance its impact, I recommend addressing the concerns regarding the animal model and noting the “non-gestating sows”.
Author Response
Comment: The manuscript presents valuable research on energy expenditure and maintenance requirements in first-parity sows. The experimental design, particularly the use of indirect calorimetry and a well-controlled feeding regimen, provides important insights into the energy metabolism of modern sows. The findings regarding the higher maintenance energy requirements (MEm) in modern breeds compared to previously reported values are relevant and timely, given recent genetic advances. The study fills an important gap in understanding the energy requirements of sows, which is essential for optimizing productivity and welfare in the swine industry. In general, I would recommend that this manuscript be published in the journal, provided the authors address the following points.
Response: The authors thank the reviewer for the detailed consideration of the manuscript. Responses are below and changes have been marked in red within the updated document.
Comment 1: The focus on first-parity sows and the detailed examination of energy expenditure at different feeding levels addresses a critical topic in swine nutrition, particularly in relation to modern genetics. One major flaw is that the current findings was obained using non-pregnant sows, yet most of the comparation was conducted against pregnant sows. It is a common concept that the pregnant sows had a greater energy utilization due to the pregnancy anobolism, thus the conclusion must be drawn carefully.
Response 1: The authors deliberately choose non-pregnant sows for this experiment to avoid confounding effects of pregnancy on energy utilization and thus maintenance requirements.
Comment 2: Methodology: The use of indirect calorimetry and frequent feeding protocols, combined with robust statistical analysis, ensures the reliability of the results. However, the use of only five (simetimes showed six) sows limits the generalizability of the findings. A larger sample size would strengthen the conclusions drawn. Additional, the number of animals was not consisitent througout the study. There are six sows described in line 43, yet only five sows were used in line 15.
Response 2: The authors agree that the limited sample size is a concern for the generalizability of the results beyond the scope of this manuscript. However, it should be noted that the results within the project were consistent with each other and support the observations for this population of sows.
Line 43 has been updated to reflect that five sows completed the experiment.
Comment 3: In the statistical analysis, I am unsure how the data were evaluated. Was a repeated measures approach used, incorporating feeding level, period, and the interaction between feeding level and period?
Response 3: The experimental model investigated the effect of ‘feeding level’ as the classification variable on the parameters of interest. Individual animals were ‘random variables’ rather than utilizing repeated measures approach. A repeated measures approach would model each animal’s response over multiple time points, focusing on within-subject changes over time. Instead, by not using the repeated measures approach, the model is more likely examining the effect of feeding level on each animal independently. Significance was taken at P < 0.05, while 0.05 < P < 0.1 was regarded as a tendency. Lines 150-161 were updated: Feeding level was treated as a fixed effect to categorize the animals based on the level of feed allowing the model to evaluate the influence on the parameters of interest (i.e., weight gain, nutrient digestibility, and energy retention). Considering animals as random variables accounts for variability between animals without attributing it to time-dependent factors.
Comment 4: In the Calculations of Energy Retention: The methodology for calculating maintenance energy requirements is solid, but the rationale for using different efficiency values (1.43 vs. 0.80) could be better explained, particularly for readers less familiar with this calculation model.
Response 4: The authors have updated Lines 145-149: Efficiencies of energy utilization often vary due to source, namely dietary sources versus body tissue. In this experiment, the value 1.43 is the reciprocal of the efficiency for utilization of ME from dietary sources where the values of 70% was reported by [12]. Alternatively, the value 0.80 was used because energy from body tissue replaces ME with a greater efficiency com-pared to dietary sources, specifically 80% [12].
Comment 5: Statistical Power: While some differences are noted as tendencies (P values between 0.05 and 0.1), these should be interpreted with caution. It may be helpful to further discuss the biological relevance of these trends.
Response 5: The authors have added “However, measurement of nutrient digestibility was not the primary focus of the trial and results should be interpreted with caution” in Lines 299-301.
Comment 6: The manuscript could benefit from a clearer presentation of the key findings. For example, the figures and tables could be more intuitive by highlighting the most important comparisons and including more concise legends if possible.
Response 6: The authors appreciate the comments; however, no changes were made to legends in order to maintain the independence of the figures and tables.
Comment 7: Also, for the Physiological States Analysis: Although the study examines various physiological states (e.g., 'nibbling', 'post-prandial'), the extrapolation of these states to 24-hour values is not entirely convincing. Additional explanation or references to justify these extrapolations would be beneficial.
Response 7: The authors have added additional explanation and justification for extrapolations in Lines 317-328: Estimating daily energy expenditure is essential for understanding metabolic needs and guiding dietary recommendations in both animal and human studies. However, directly measuring energy expenditure over a full 24-hour period can be challenging (Bell et al, 1986), time-consuming, and costly (Szwiega et al., 2023). To mitigate this, researchers often extrapolate shorter measurement intervals to estimate total daily energy expenditure, though this approach requires careful methodology to ensure accuracy. Statistical modeling and regression techniques have been successfully used to extrapolate shorter measurements by adjusting for specific time-dependent factors, such as rest versus activity periods (Ravussin & Bogardus, 1989). These approaches provide a more reliable estimate by accounting for metabolic fluctuations that occur within a day. In livestock management, where continuous monitoring is impractical, extrapolating shorter measurement periods with these models offers a feasible and efficient method to estimate daily energy needs, aiding in more accurate feeding strategies (Whittemore et al., 2001).
Ravussin, E., & Bogardus, C. (1989). Relationship of genetics, age, and physical fitness to daily energy expenditure and fuel utilization. *American Journal of Clinical Nutrition*, 49(5), 968-975.
Whittemore, C. T., Green, D. M., & Knap, P. W. (2001). Technical note: A stochastic model for predicting feed intake of growing pigs. *Journal of Animal Science*, 81(5), 1482-1488.
Bell EF, Rios GR, Wilmoth PK. Estimation of 24-hour energy expenditure from shorter measurement periods in premature infants. Pediatr Res. 1986 Jul;20(7):646-9. doi: 10.1203/00006450-198607000-00013. PMID: 3725462.
Szwiega S, Pencharz PB, Ball RO, Tomlinson C, Elango R, Courtney-Martin G. Amino acid oxidation methods to determine amino acid requirements: do we require lengthy adaptation periods? Br J Nutr. 2023 Jun 14;129(11):1848-1854. doi: 10.1017/S0007114522002720. Epub 2022 Sep 1. PMID: 36045125; PMCID: PMC10167660.
Comment 8: Terminology: There are instances where more precise language could improve readability, especially when describing metabolic rates and energy turnover.
Response 8: The authors have reviewed the paper and attempted to identify and make improvements.
Comment 9: The discussion section should be improved.
Response 9: The authors have reviewed the paper and attempted to identify and make improvements.
Comment 10: Lines 263-264, how MEm been determined by “using the formulae of 234 Lodge [11] (Table 5)”.
Response 10: The authors have updated these lines: The determined MEm for each individual non-pregnant first-parity animal within this experiment was calculated using the formulae of Lodge [11]. The formulae were presented in Lines 134-144.
Comment 11: Line 237, the ref. [14] was based on weaning piglets, could be suitable for elucidating this issue?
Response 11: The authors did not encounter a more relevant reference involving pigs fed different planes of nutrition and calculation of MEm. Line 240 has been updated with additional information.
Comment 12: Line 240, indicates the animal (growing pig) when citing the ref. [15].
Response 12: The authors did not encounter a reference involving sows.
Comment 13: Line 262-263, I do not agree with the notion that a negative in energy retention (-0.1MJ/d) is consistent with a negative weight gain (-198 ± 96 g/d), that does not make sense. The RQ was above 1.0 (1.03) for the 1.0 time in table 3, indicating lipid retention. How to explain this?
Response 13: The authors determined the RQ values to be not different than 1 (data not shown). Furthermore, the difference (-1.6 kg) was not significant (P = 0.11), suggesting that sows did not significantly change body weight.
Comment 14: I would suggest the authors discussion why the ADF and organic matter greater in 2.0 times than the 1.0 times.
Response 14: The authors understand this to be referencing the results presented in Table 2 where the ADF digestibility was greater and organic matter tended to be greater when sows were fed 2.0 times maintenance versus sows fed 1.0 times maintenance. The authors note that Chassé et al., 2023 observed that larger meals (2.0 times maintenance) increased feed retention time in the gastrointestinal tract potentially due to decreased motility and slower gastric emptying and would allow more time for microbial fermentation compared to smaller meals (1.0 time maintenance).
Lines 295-301 have been added: The observed increase in the ATTD digestibility of ADF and tendency of increased ATTD of N and OM may due the fact that larger meals, such as feeding 2.0 times maintenance, increased feed retention time in the gastrointestinal tract potentially due to de-creased motility and slower gastric emptying and would allow more time for microbial fermentation compared to smaller meals (1.0 time maintenance) [28]. However, measurement of nutrient digestibility was not the primary focus of the trial and results should be interpreted with caution.
Comment 15: Conclusion: This manuscript contributes significantly to the understanding of energy metabolism in modern sows. However, to enhance its impact, I recommend addressing the concerns regarding the animal model and noting the “non-gestating sows”.
Response 15: The authors have updated the Conclusion by adding “non-pregnant” in Line 343.
Round 2
Reviewer 1 Report
Comments and Suggestions for Authors
The ceoncerns of this reviewer has been addressed.